# Use of Light Spectra for Efficient Production of PLBs in Temperate Terrestrial Orchids

**Hossein Naderi Boldaji, Shirin Dianati Daylami * and Kourosh Vahdati**

Department of Horticulture, College of Aburaihan, University of Tehran, Tehran 3391653755, Iran;
hossein.naderi87@ut.ac.ir (H.N.B.); kvahdati@ut.ac.ir (K.V.)
*  Correspondence: dianati@ut.ac.ir; Tel.: +98-9126359782

**Abstract:** Wild orchids, especially the terrestrial temperate ones are endangered species due to challenges in their natural habitats. Therefore, there is an urgent need to introduce efficient propagation methods to overcome the natural reproduction problems of these orchids. In this study, the effects of different light spectrums, explant types, wounding, and combinations of different plant growth regulators (PGRs) on direct somatic embryogenesis (DSE) of two species of these endangered orchids listed in the conservation category, were studied. The highest percentages of DSE formation and embryo germination were observed in *Dactylorhiza umberosa* protocorm explants exposed to white light (400–730 nm) and in *Epipactis veratifolia* protocorm explants exposed to a combination of red and far-red spectra (R: FR = 70:30). This occurred while red (610–700) alone and in combination with far-red (710–730 nm) spectrum induced embryogenesis more than the blue spectrum and dark condition in *E. veratifolia*. Thidiazuron (TDZ, 3 mg L$^{-1}$), produced the highest percentage of protocorm-like bodies (PLBs) on protocorm explants in both orchids. Kinetin (Kin, 2 mg L$^{-1}$) and Benzyladenine (BA 3 mg L$^{-1}$) had the most effect on the survival and growth of PLBs, respectively, in *D. umberosa* and *E. veratifolia*. Species did not show similar embryogenesis responses under light spectrums. In a medium containing 3 mg L$^{-1}$ TDZ, white light and R-FR spectra produced the most PLBs on wounded protocorm explants of *D. umberosa* and *E. veratifolia* respectively. The developmental stage of apical meristem of PLBs in both species was more advanced under R-B spectra, compared to others.

**Keywords:** *Epipactis veratifolia*; *Dactylorhiza umberosa*; direct somatic embryogenesis; plant growth regulator; protocorm-like body; wounding

## 1. Introduction

Orchids are high-price ornamental crops that have attracted the attention of their production as pot plants and cut flowers [1]. Of the wild orchids, native orchids such as saleps have traditionally been harvested from their natural habitats for their use in the medicinal and food industries. The tubers of some terrestrial orchids like *Orchis mascula*, *Dactylorhiza umberosa*, and some other species are called salep [2]. Salep also refers to the powder of dried tubers that are used in the production of ice cream, drinks, medicines, confectionery, and hot drinks [3]. The scarcity of resources for salep has led to the uncontrolled harvesting of these plants from their natural habitats [4], making them an endangered species. *Epipactis veratrifolia* is one of the wild orchids used in traditional medicine and has a high potential for breeding as an ornamental plant [5]. Low seed germination, slow vegetative growth, and lack of proper coexistence have sometimes made their propagation by conventional methods impossible [6,7]. To achieve successful regeneration, in vitro techniques are now efficiently practiced and used for sexual and non-sexual propagation of orchids, especially for endangered orchid species [8,9]. There are various methods for mass in vitro culture of orchids such as culture of seeds [10], shoot tip and axillary bud culture [11], and using protocorm-like bodies (PLBs) [12], and flower buds [13]. Direct and indirect

somatic embryogenesis (DSE), (ISE) and protocols have been offered for the induction of protocorm-like bodies (PLBs) in orchids [5–7]. These micropropagation methods have been used for somatic embryogenesis (SE) in many orchids such as *Cymbidium* [14], *Oncidium* [15], *Phalaenopsis* [1], and *Xenikophyton smeeanum* [16], however, there is not a wealth of information for in vitro micropropagation of temperate terrestrial orchids [17]. Explant type, genotype, medium composition, plant growth regulators (PGRs), and light regime can be mentioned as factors influencing somatic embryogenesis (SE) in Oncidium orchids [18]. Light is a determinant factor for the tissue culture of plants. Different light spectra have been used to study their effects on plant growth and organogenesis [19–21]. Light-emitting diodes (LEDs) have emerged as a new light source for in vitro culture. Different kinds of artificial light can play a key role in successful in vitro plant production, besides other factors such as gas exchange in the culture vessel, temperature, and composition of the culture medium [22]. The effects of the light spectrum have been investigated on the in vitro growth of many plant species such as *Lilium* 'Pesaro' [23], *Dianthus caryophyllus* [24], and *Zantedeschia jucunda* [25]. The emission of light in LED lighting systems allows the selection of spectra quality and provides the opportunity for the regulation of photosynthetic and photomorphogenic reactions required for an in vitro culture of plants [26]. Significant improvements have been achieved in increasing the fresh and dry weight of shoots and proliferation rate by changing the photoperiod regime from 16 h to a 4 h photoperiod, thereby allowing explants to do a better exchange of $CO_2$ [27,28]. Furthermore, the light has been introduced as one of the important inducers for the generation of SE [29]. Effects of various spectra of LED lights have been reported in some orchids such as *Oncidium* [18], *Phalaenopsis* [30], and *Cymbidium* [31], as well as other plant species such as *China Rose* [32], and *Agave tequilana* [33]. The induction of SE and callus was obtained under different light conditions in different plant species. For instance, red, red + far-red lights in *Phalaenopsis*, red light in *Cymbidium*, red and white light in *Agave tequilana,* and red light in *Rosa chinensis* [30–33]. PGRs are among the most important factors that influence SE. Auxins and cytokinins can be mentioned specifically as the most used ones [34]. Applying PGRs can effectively improve DSE or ISE [24]. Positive effects of PGRs on embryogenesis have also been reported in plant species such as *Phalaenopsis amabilis* [34], *Anthurium* [35], *Lilium ledebourii* [36], and *Cyclamen* [37]. It was reported that wounding of the explants can lead to faster SE induction in soybeans [38], while there is little information on the effects of wounding on embryogenesis in plants such as orchids [39]. Most studies have been limited to tropical orchids, and only a few studies have been conducted on SE and regeneration in temperate terrestrial orchids. To the best of our knowledge, there are no studies on the effects of light spectra and wounding on somatic embryogenesis and regeneration in terrestrial orchids. In the present study, we have successfully established an in vitro propagation protocol for *D. umberosa* and *E. veratifolia* by proliferation through PLBs to facilitate the conservation, cultivation, and introduction of these plants into future flower markets.

## 2. Materials and Methods

### 2.1. Plant Materials and Explant Preparation

In this study, two native Iranian orchids were utilized; namely the rhizomatous species, *E. veratrifolia* and the tuberous species *D. umberosa*. Capsules containing mature brown seeds of these plants were collected from their natural habitats in the Alborz Mountains. Capsules were sterilized for 20 min in 20% sodium hypochlorite (NaOCl), followed by 3 rinses with sterilized distilled water; then about 100 tiny, dusty seeds (There are more than a thousand seeds in each capsule) were cultured on modified FAST (MFAST) as a basal medium in each Petri dish [40]. Four types of explants were used. Protocorm was the first kind of explant. At the end of the germination process of orchid seeds, embryos form small spherical tuber-like bodies referred to as protocorms and were used in two forms: wounded and non-wounded in both species (Figure 1f). After seedling growth, three other types of explants were prepared, which included leaf segment (in both species Figure 1d), single node (in *E. veratrifolia*-Figure 1e), and crown of the plantlet (in *D. umberosa*). The

explants were cultured on a basal medium, containing different concentrations of PGRs (including TDZ, NAA, BA, and Kin) according to Table 1.

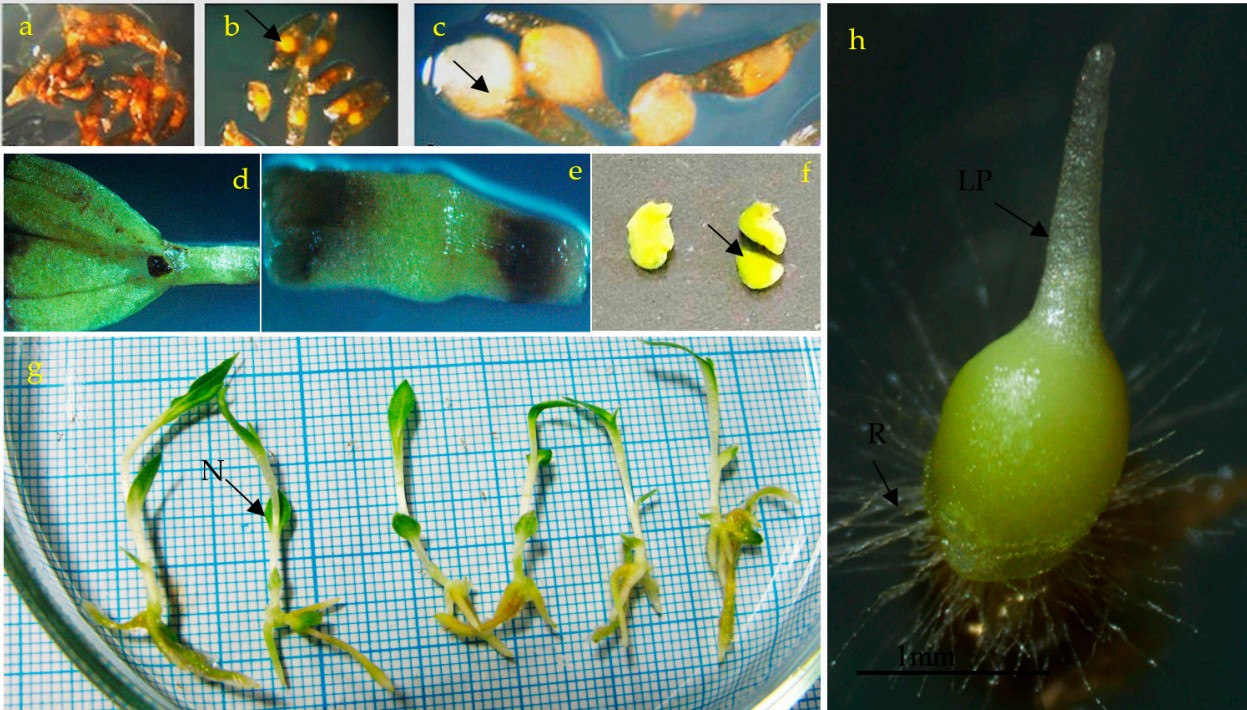

**Figure 1.** The seed germination stage of Epipactis veratifolia; kinds of explants and seedlings (**a**) cultivated seeds, (**b**) swelling of the embryos (arrow) in the first stage of germination, (**c**) rupture of seed testa by swelled embryos, (**d**) leaf explant (basal part of leaf), (**e**) stem node explant, (**f**) intact and wounded protocorm explants; the end part without shoot apical meristem (arrow) used as explants, (**g**) in vitro rooted seedlings; their leaf and stem nodes (N) used as explants, (**h**) a protocorm resulting from the development of a Dactylorhiza umberosa seed embryo at the stage of converting apical meristem (AP) to leaf primotdia (LP), with many hairy rhizoids (R) at the bottom end.

**Table 1.** Effects of PGRs and explant types on the mean formation of somatic embryos (PLBs) and their germination rate, in *Epipactis veratifolia and Dactylorhiza umberosa.* (×) in column Shows the kinds of explant. Within each column, different letters indicate significant differences at $p < 0.05$ using ANOVA and Duncan's multiple range tests.

| Plant | Concentration of PGRs (mg/L) | | | | Explant Type | | | | PLBs Formation (%) | PLBs Growth (%) |
|---|---|---|---|---|---|---|---|---|---|---|
| | TDZ | NAA | BA | Kin | Protocorm | Leaf | Node | Crown | | |
| | 0 | 0 | 0 | 0 | × | - | - | - | 0 | 0 |
| | 3 | 0 | 0 | 0 | × | - | - | - | 100 [a] | 100 [a] |
| | 0 | 0 | 1 | 0 | - | × | - | - | 0 | 0 |
| | 0 | 0 | 1 | 0 | - | - | × | - | 0 | 0 |
| | 0 | 0 | 2 | 0 | - | × | - | - | 0 | 0 |
| *tiEpipactis verafolia* | 0 | 0 | 2 | 0 | - | - | × | - | 0 | 0 |
| | 0 | 0 | 3 | 0 | - | × | - | - | 0 | 0 |
| | 0 | 0 | 3 | 0 | - | - | × | - | 0 | 0 |
| | 0 | 0.5 | 0 | 0 | - | × | - | - | 0 | 0 |
| | 0 | 0.5 | 0 | 0 | - | - | × | - | 25 [b] | 25 [d] |
| | 0 | 0.5 | 1 | 0 | - | × | - | - | 0 | 0 |
| | 0 | 0.5 | 1 | 0 | - | - | × | - | 25 [b] | 75 [b] |
| | 0 | 0.5 | 2 | 0 | - | × | - | - | 0 | 0 |

**Table 1.** *Cont.*

| Plant | Concentration of PGRs (mg/L) | | | | Explant Type | | | | PLBs Formation (%) | PLBs Growth (%) |
|---|---|---|---|---|---|---|---|---|---|---|
| | TDZ | NAA | BA | Kin | Protocorm | Leaf | Node | Crown | | |
| | 0 | 0.5 | 2 | 0 | - | - | × | - | 0 | 0 |
| | 0 | 0.5 | 3 | 0 | - | × | - | - | 0 | 0 |
| | 0 | 0.5 | 3 | 0 | - | - | × | - | 0 | 0 |
| | 0 | 1 | 0 | 0 | - | × | - | - | 0 | 0 |
| | 0 | 1 | 0 | 0 | - | - | × | - | 38 [a] | 20 [d] |
| | 0 | 1 | 1 | 0 | - | × | - | - | 0 | 0 |
| | 0 | 1 | 1 | 0 | - | - | × | - | 25 [b] | 50 [c] |
| | 0 | 1 | 2 | 0 | - | × | - | - | 0 | 0 |
| | 0 | 1 | 2 | 0 | - | - | × | - | 18 [c] | 75 [b] |
| | 0 | 1 | 3 | 0 | - | × | - | - | 0 | 0 |
| | 0 | 1 | 3 | 0 | - | - | × | - | 0 | 0 |
| | 0 | 0 | 1 | 0 | - | × | - | - | 0 | 0 |
| | 0 | 0 | 1 | 0 | - | - | × | - | 0 | 0 |
| | 0 | 0 | 2 | 0 | - | × | - | - | 0 | 0 |
| | 0 | 0 | 2 | 0 | - | - | × | - | 25 [b] | 50 [c] |
| | 0 | 0 | 3 | 0 | - | × | - | - | 0 | 0 |
| | 0 | 0 | 3 | 0 | - | - | × | - | 13 [c] | 100 [a] |
| | 0 | 0 | 0 | 0 | × | - | - | - | 0 | 0 |
| | 3 | 0 | 0 | 0 | × | - | - | - | 100 [a] | 90 [b] |
| | 0 | 0 | 0 | 1 | × | - | - | - | 0 | 0 |
| *Dactylorhiza umberosa* | 0 | 0 | 0 | 1.5 | × | - | - | - | 25 [c] | 75 [b] |
| | 0 | 0 | 0 | 2 | × | - | - | - | 50 [b] | 100 [a] |
| | 2 | 0 | 2 | 0 | - | - | - | × | 0 | 0 |
| | 1 | 0.5 | 0 | 0 | - | - | - | × | 0 | 0 |
| | 0 | 0.5 | 2 | 0 | - | - | - | × | 0 | 0 |

## 2.2. Media and Culture Conditions

Cultivated seeds (Figure 1) were grown using a hormone-free modified FAST medium (MFAST) as the basal medium containing both macro and micro elements (Merck, Hesse-Darmstadt, Germany). The medium was supplemented with myo-inositol 100 mg $L^{-1}$, nicotinic acid 0.5 mg $L^{-1}$, pyridoxine HCl 0.5 mg $L^{-1}$, thiamine HCl 0.1 mg $L^{-1}$, glycine 2 mg $L^{-1}$, sucrose 3 g $L^{-1}$, peptone 2% and agar 4.8 g $L^{-1}$. [40]. The pH was adjusted to 5.5 ± 0.1 and the solution was autoclaved at 121 °C for 20 min. The same medium was used for continuing growth of explants for a period of 6 to 8 weeks. To study the effects of four kinds of growth regulators (PGRs), including thidiazuron (Sigma Aldrich, UAS) (TDZ 0, 2 and 3 mg $L^{-1}$), N6-benzyleadenine (BA 0, 1, 2 and 3 mg $L^{-1}$), ∞-naphthaleneacetic acid (NAA 0, 0.5 and 1 mg $L^{-1}$), kinetin (KIN 0, 1, 2 and 3 mg $L^{-1}$) and four kinds of explants (including five explants of each type of crown, node, leaf segment, and protocorm) were cultured in separate Petri dishes (five micro-samples in each of three replicates) and placed in a growth room under white LED lights (Figure 1). Then the interaction between TDZ (0 and 3 mg $L^{-1}$), wounding, and light spectra was investigated on SE of protocorm explants (wounded or un-wounded) in both species. The light treatments including white light (W, as the control range at 400–700 nm), blue (B) range at 460 nm, red (R) range at 660 nm, green (G) range at 530 nm, combination of red and blue (R:B = 70:30), also a combination of red and far-red (R:FR = 70:30) were provided using LED lamps at light intensity of 80 μmol m$^{-2}$ s$^{-1}$ as well as darkness condition, as the control. The cultures were placed in a growth room with a temperature of 22 ± 2 °C, a light period of 16 h and, a relative humidity of 70%. Wavelengths were measured using a Sekonic C7000 spectrometer (Sekonic Corp., Tokyo, Japan) within the range of 300–800 nm. After four weeks, induction

of DSE (globular type) was evaluated using a microscope. Following SE induction, the cultures were transferred to the hormone-free MFAST medium and placed under the light with the same environmental conditions as previously described.

### 2.3. Germination and Acclimatization

Embryos produced via DSE were counted as germinated (Table 1) and transferred to a hormone-free basal medium for plant growth under the environmental conditions described above. For acclimatization, the plantlets (developed from DSE) were transferred to plastic pots filled with a sterile coco-peat: perlite mixture (3:1), and placed in an adapted chamber at a temperature of $28 \pm 2$ °C.

### 2.4. Statistical Analysis

Experiments were arranged as factorial in a completely randomized design. The data were subjected to analysis of variance (ANOVA) and means were compared using Duncan's multiple range tests at $p < 0.05$ probability level using the SAS 9.3 software.

## 3. Results

### 3.1. Effects of PGRs, Explant Type, and Wounding on DSE

Embryo formation was observed three weeks after placing the explants in the growth medium. The results of the statistical analysis showed a significant difference between the interaction of treatments (growth regulators, the type of explant, and the light spectrum). The percentage of DSE formation, PLBs germination, and final plantlets formation, especially on protocorm explants, in both species, had significant differences ($p < 0.05$) under various light spectra (Figures 2–8). The type of PGR caused a significant effect on embryo formation in both species (Table 1, Figure 4). Between the PGR treatments, *E. veratifolia* showed the highest embryogenesis frequency (100%) when grown on a medium supplemented with 3 mg L$^{-1}$ TDZ, using protocorms and followed by 1 mg L$^{-1}$ NAA (38%) in node explants (Table 1). The medium supplemented with 3 mg L$^{-1}$ BA had the lowest SE rate at 13%. Table 1 shows an increase in DSE production with increasing NAA concentration (from 0 to 1 mg L$^{-1}$) in combination with BA. NAA treatment (0.5 mg L$^{-1}$) alone and in combination with 1 mg L$^{-1}$ BA, caused 25% PLB formation on the node explants in *Epipactis veratifolia*, but in combination with BA, the survival of these somatic embryos (PLBs growth %) was 50% more (Table 1). The positive effect of BA on the survival of somatic embryos is also observed in combination with 1 mg L$^{-1}$ of NAA (Table 1).

Different concentrations of the BA alone or in combination with NAA did not affect the leaf explants of *Epipactis veratifolia* (Table 1). Three weeks after placing the explants in the growth medium, somatic embryos formed on protocorm explants of *D. umberosa*, under all light spectra. Embryo formation and germination rate, percentage of DSE, embryo germination, and plantlet production were significantly ($p < 0.05$) influenced by the type of explant, the combination of PGRs, light spectrums, and wounding (Table 1, Figures 4–6 and 8). The kind of PGRs utilized had a considerable effect on the DSE percentage (Table 1). Only protocorm explants that developed globular embryos displayed somatic embryo initiation in *D. umberosa* (Table 1). The DSE was induced on the protocorm explants after three to four weeks. The maximum rate (100%) of embryo formation (DSE) was observed in a medium containing 3 mg L$^{-1}$ TDZ (Table 1). In contrast, no embryos were observed on crown explants of *D. umberosa* in all PGR treatments. Kin in 1.5 and 2 mg L$^{-1}$ concentrations, also caused 25% and 50% direct embryogenesis in protocorm explants of *D. umberosa*, respectively. All concentrations of PGRs (TDZ, BA, and NAA) did not affect the crown explant. SE was observed on both protocorm (100%) and stem node (38%) explants of *E. veratifolia* (Table 1). R-FR spectra caused most DSE on non-wounded protocorm explants of *E. veratifolia* (Figures 2 and 5a). In *D. umberosa* only one form of the explant (protocorm) produced a number of somatic embryos (Figure 3) with a high frequency of embryo germination (Table 1). Therefore, the protocorm explant was selected to proceed with the

experiment. The most somatic embryo of *D. umberosa* was obtained under white light and wounding, which had a positive effect on the DSE of this species (Figure 5b).

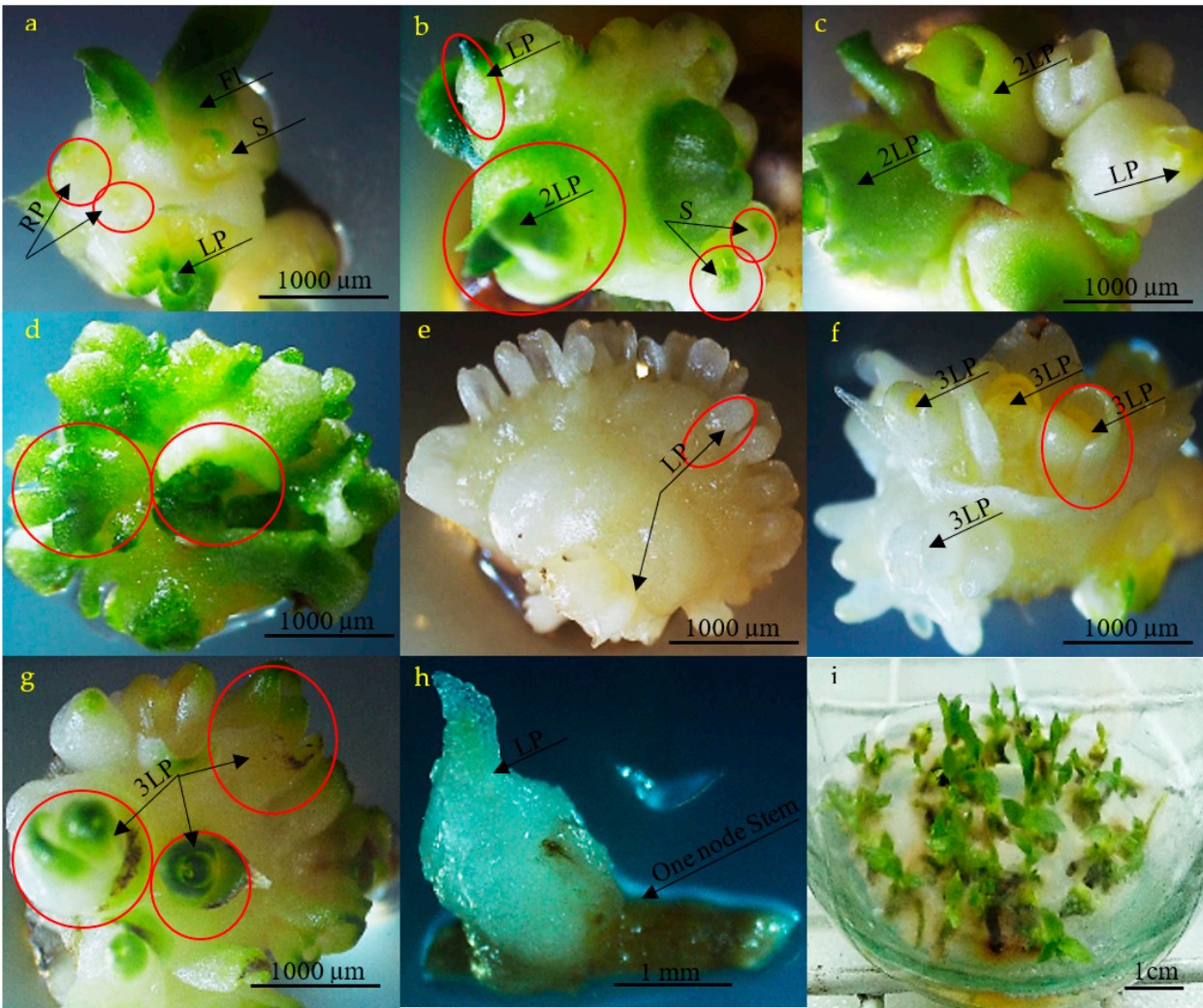

**Figure 2.** PLBs formation of Epipactis veratifolia after direct somatic embryogenesis under different light spectra; on protocorm explants ((**a**–**g**). in medium contain 3 mg L$^{-1}$ TDZ) and node explant ((**h**), in medium contain 0.5 mg L$^{-1}$ NAA); (**a**) Cluster of PLBs at different developmental stages, including rudimentary PLBs (RP), PLBs with shoot apical meristem (SM), PLBs with first leaf primordia (LP) and PLBs with first leaf (FL) under blue spectrum. Each red circle represents a PLB which its stage or morphology is explained by arrow, (**b**) PLBs with huge diameter under green spectrum, (**c**) large PLBs with two leaf primordia (2LP) under red + blue spectra, (**d**) deformed PLBs with branched first leaf primordia under red spectrum, (**e**) many tiny PLBs with one leaf primordia without chlorophyll under red + far spectra, (**f**) PLBs with three leaf primordia (3LP) without chlorophyll under dark conditions, (**g**) many PLBs with three leaf primordia (3LP) under white light, (**h**) direct somatic embryo (one PLB) formation on node explant under dark condition by using 0.5 mg L$^{-1}$ NAA, (**i**) growth of somatic plantlet on hormone-free medium.

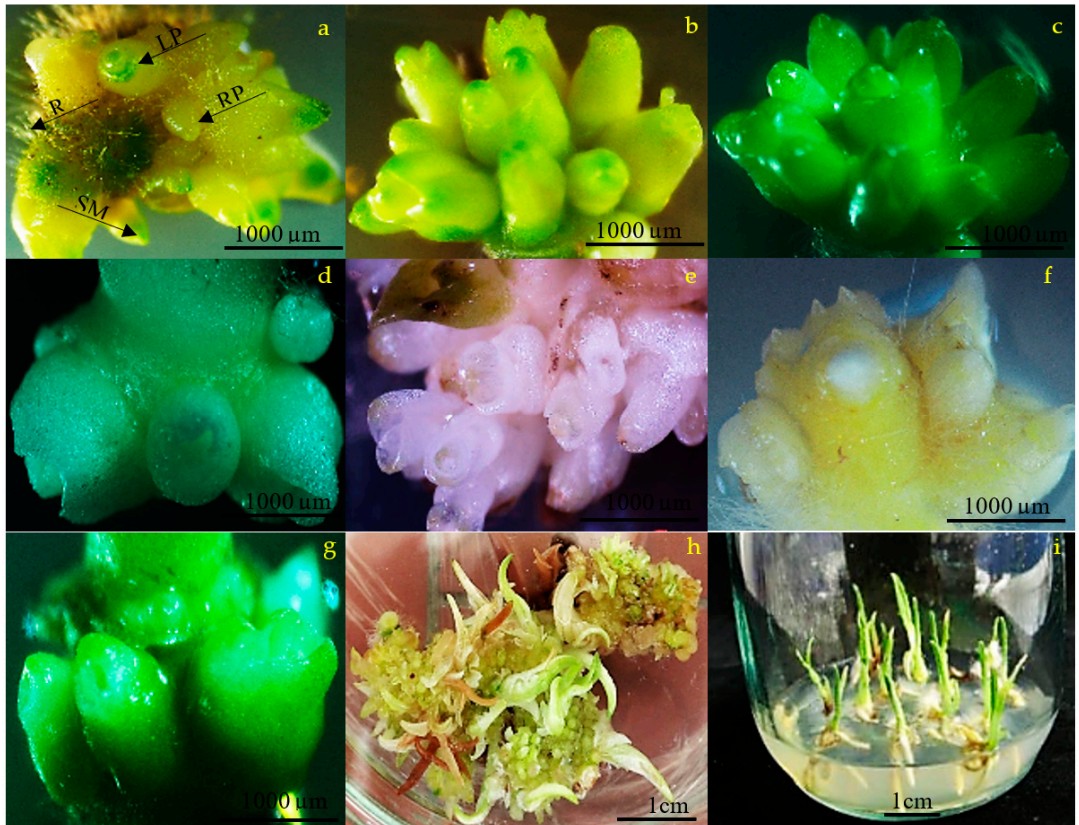

**Figure 3.** Direct somatic embryogenesis (formation of PLBs) on protocorm explants of Dactylorhiza umberosa under different light conditions including; (**a**) Cluster of PLBs at different stages of development stages with many rhizoids (R) under blue spectrum (in media containing 1.5 mg L$^{-1}$ Kin), (**b**) under white light (in media containing 2 mg L$^{-1}$ Kin), (**c**) red + far-red spectra (in media containing 3 mg L$^{-1}$ TDZ), (**d**) dark condition (in media containing 1.5 mg L$^{-1}$ Kin), (**e**) red + blue spectra (in media containing 3 mg L$^{-1}$ TDZ), (**f**) green spectra (in media containing 3 mg L$^{-1}$ TDZ), (**g**) red spectra (in media contain 3 mg L$^{-1}$ TDZ), (**h**) germination of somatic embryos (PLBs), (**i**) growth of somatic plantlets.

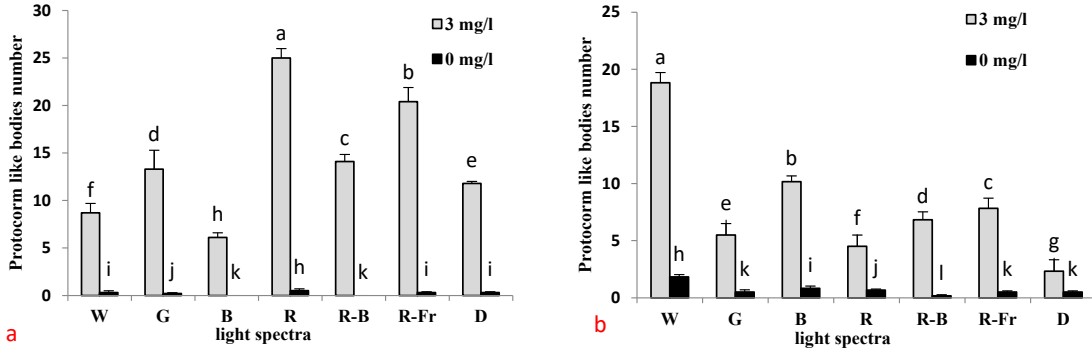

**Figure 4.** Interaction effects of TDZ concentrations (0 and 3 mg L$^{-1}$) and different light spectra on DSE of protocorm explants of Epipactis Veratifolia (**a**) and Dactylorhiza umberosa (**b**). The horizontal axis shows the light treatment (W = white, G = green, R = red, R-B = red + blue, RFR = red + far-red, and D = dark). The columns show somatic embryogenesis (the mean number of embryos (PLBs) per treatment). Values are the mean of three replicates and bars represent the standard errors. Data were recorded two months after embryo formation. Different letters indicate significant differences at $p < 0.05$ using Duncan's multiple range test.

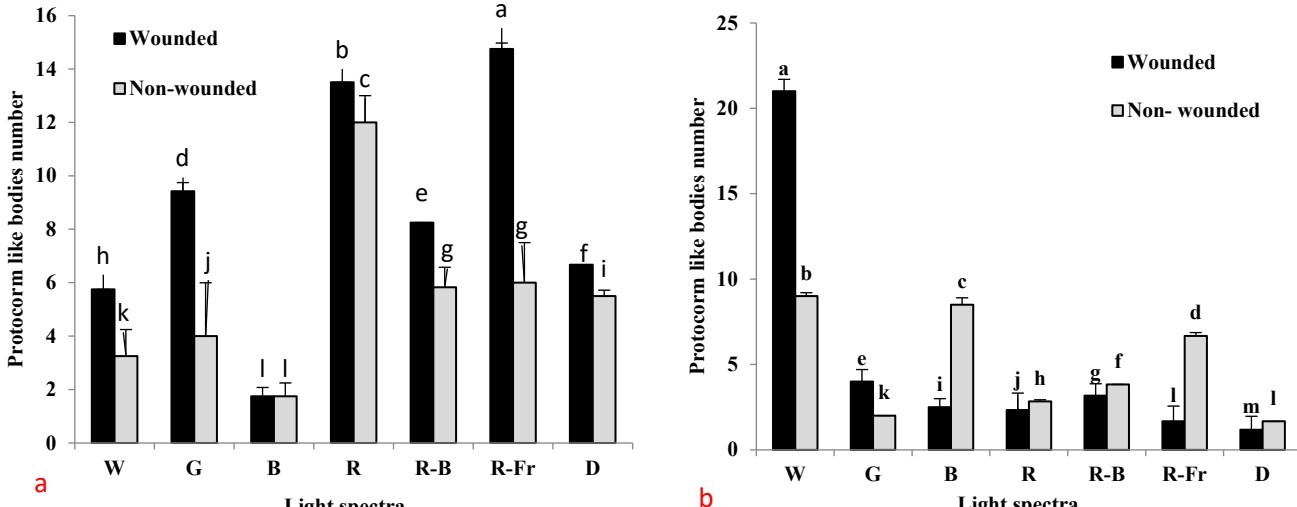

**Figure 5.** Interaction between different light spectra and wounding on embryogenesis (PLBs formation on protocorm explants) in Epipactis veratifolia (**a**) and Dactylorhiza umberosa (**b**) on media containing 3 mg L$^{-1}$ TDZ. The horizontal axis shows the light treatment (W = white, G = green, R = red, R-B = red + blue, RFR = red + far-red, and D = dark). The vertical columns show the means of somatic embryogenesis with wounding/unwounding explants. Values are the means of three replicates and bars represent the standard errors. Data were recorded two months after the formation of embryos Different letters indicate significant differences at $p < 0.05$ using Duncan's multiple range test.

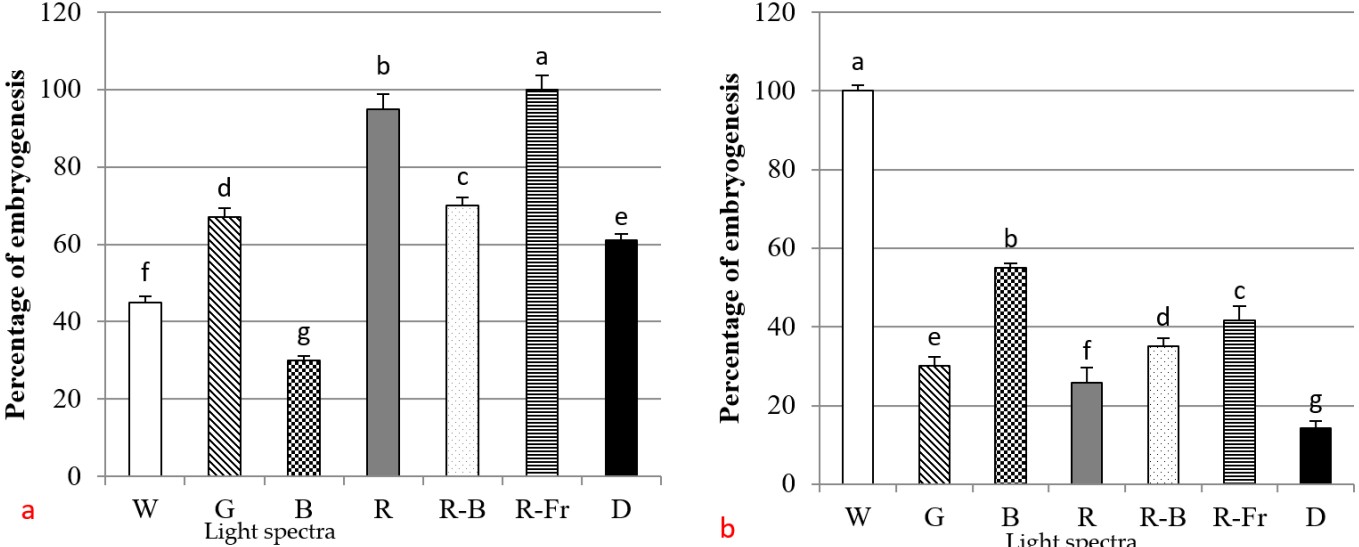

**Figure 6.** Influence of different light spectra on the induction of somatic embryos% on protocorm wounded explants from *Epipactis veratifolia* (**a**) and *Dactylorhiza umberosa* (**b**) in media containing TDZ. The horizontal axis shows the light treatment (W = white, G = green, R = red, R-B = red + blue, RFR = red + far-red, and D = dark). The vertical columns show the means of somatic embryogenesis under the wounding/unwounding and PGR-free/TDZ conditions. Values are the means of three replicates and bars represent the standard errors. Different letters indicate significant differences at $p < 0.05$ using Duncan's multiple range test.

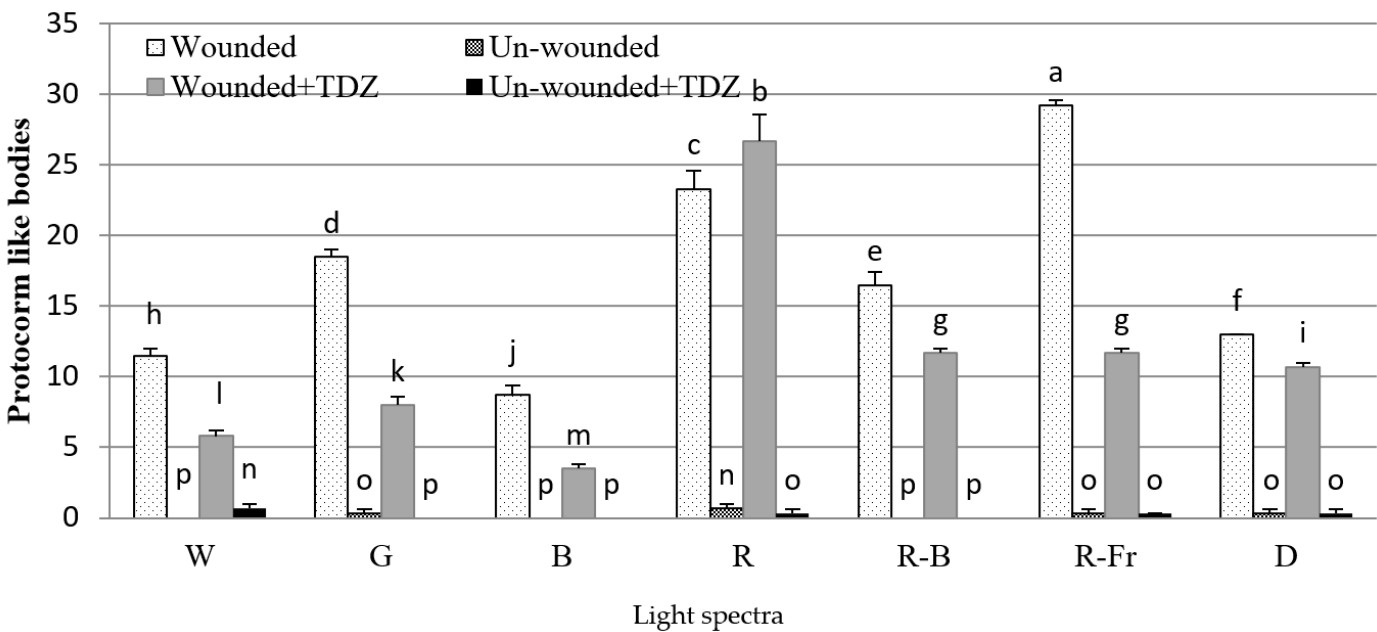

**Figure 7.** Interaction of different light spectra, TDZ application, and wounding on the induction of somatic embryos (PLBs) in Epipactis Veratifolia. The horizontal axis shows the light treatment (W = white, G = green, R = red, R-B = red + blue, RFR = red + far-red, and D = dark). The vertical columns show the means of somatic embryogenesis under the wounding/unwounding and PGR-free/TDZ conditions. Values are the means of three replicates and bars represent the standard errors. Different letters indicate significant differences at $p < 0.05$ using Duncan's multiple range test.

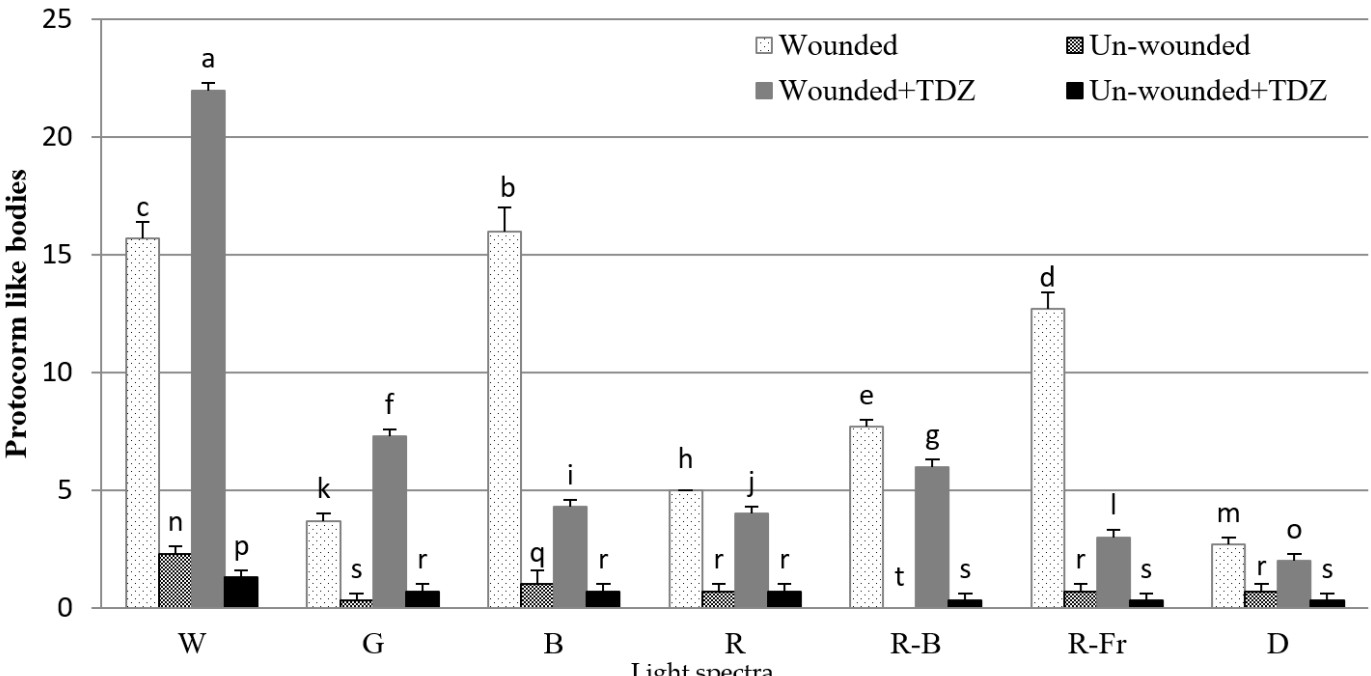

**Figure 8.** Interaction of different light spectra, TDZ application, and wounding on the induction of somatic embryos (PLBs) in Dactylorhiza umberosa. The horizontal axis shows the light treatment (W = white, G = green, R = red, R-B = red + blue, RFR = red + far-red, and D = dark). The vertical columns show the means of somatic embryogenesis under the wounding/unwounding and PGR-free/TDZ conditions. Values are the means of three replicates and bars represent the standard errors. Different letters indicate significant differences at $p < 0.05$ using Duncan's multiple range test.

### 3.2. Induction of SE under Different Light Spectra

Different light spectrums had a notable impact on DSE in both orchid types. Red light led to a higher percentage (100%) of DSE and a number of PLBs (25 PLB) in *E. veratifolia* were observed under red light (Figure 4). The percentages of DSE were also affected by the light spectrums ($p < 0.05$). The greatest DSE percentages were observed under R: FR (100%), R (95%), RB (70%), and G (67%) spectra, with the lowest DSE seen under B (30%), W (45%), and darkness (61%), respectively (Figure 6a). FR (100%), R (95%), RB (70%), and G (67%) spectra, with the lowest DSE seen under B (30%), W (45%), and darkness (61%), respectively (Figure 6a). In *D. umberosa*, W light induced the highest DSE (100%), followed by B (55%), R: FR (41.7%), RB (35%), G (30%), R (25.8%), and D (14.2%) (Figure 6b). The size of SEs was significantly larger when explants of *E. veratifolia* were exposed to B, R-B, and G spectra. Both white light and RB spectra led to greater development of PLBs with three-leaf primordia (Figure 2). Bigger PLBs of *D. umberosa* were observed under the G spectrum and dark conditions. RB spectra resulted in the development of more PLBs with three-leaf primordia in this species (Figure 3). Explants of *E. veratifolia* turned brown and subsequently exhibited necrosis when exposed to RB, W, and R lights. As a result, the survival rate of plantlets decreased (Figure 9). Furthermore, the light spectrum had an effect on embryo germination in both orchids. The embryos generated under R-Fr and W lights resulted in the highest percentage of embryo germination and plantlet development in *E. veratifolia* (100%) and *D. umberosa* (100%), respectively (not shown).

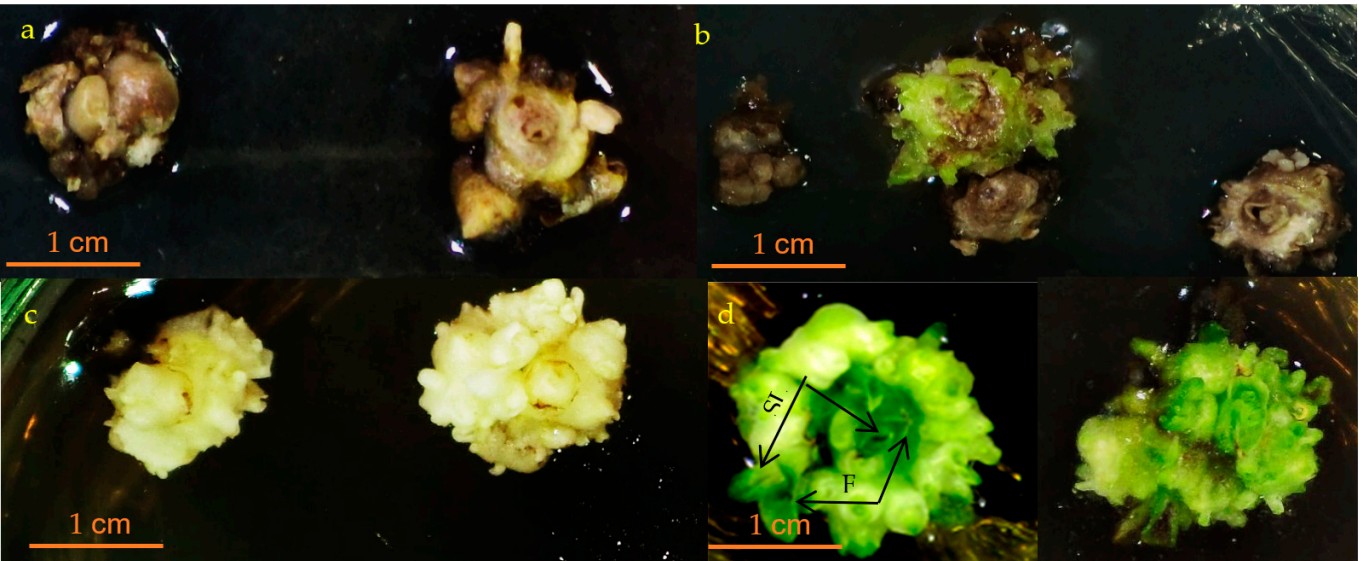

**Figure 9.** Growth and survival of PLBs after direct somatic embryogenesis in Epipactis veratifolia. (**a**) Production of the cluster with a number of large PLBs which became brown and necrotic under a blue and white LED light spectra. (**b**) Red + blue and green light spectra produced the cluster with a few numbers of PLBs that also turned brown and necrotic. (**c**) Red light produced the highest embryogenesis and PLBs proliferation rate. (**d**) Significantly larger PLBs with a higher developmental stage were observed in the cluster when explants were exposed to blue and red-blue light. In most of the PLBs, the first leaflet (FL) and in some others the second leaflet (SL) were visible under these spectra (arrow).

## 4. Discussion

The effect of different light spectra on the direct embryogenesis of temperate terrestrial native orchids and their combination with plant growth regulators, explant type, and wounding has not been previously investigated. Some researchers have studied the embryogenesis of native orchids, for example, an in vitro somatic embryogenesis method and regeneration for *Anoectochilus elatus* Lindley, an endangered orchid, by studying some PGRs and vitamins [7]. Similar methods have been developed for some of the other endemic

terrestrial orchids such as *Anoectochilus elatus* and *Caladenia latifolia* [41]. A number of studies conducted in this field on orchids have used different types of explants for DSE; for example, stem nodes [5,6,42], protocorm [41–43], apical buds, crowns, and leaves [5]. In the current study, we focused on introducing an efficient method for DSE in *D. umberosa* and *E. veratifolia* using a novel combination of light spectra, PGRs, and wounding. Leaf explants of *E. veratifolia* did not produce any SE when placed in different media with different PGRs, but protocorm and nodal explants efficiently induced SE (Table 1 and Figure 2a–h). Wounding had effects on the genes encoding the induction of cell-wall proteins [44]. In the current research, we found that in *D. umberosa*, wounded protocorm explants efficiently induced DSE compared to non-wounded ones especially under white light and G spectrum (Figure 5b). This phenomenon is observed in *E. veratifolia* under all light conditions (Figure 5a). This finding is consistent with previous findings in soybean [38] and in tomato shoot regeneration by wounding of cotyledonary explants [45]. The results suggest that intact explants are more efficient for embryo formation in *D. umberosa* under B, R, RB, RFR spectra, and dark conditions (Figure 5b). It has been previously reported that the wounding of explants causes browning, which increases the phenolic compounds and eventually leads to the death of explants [46]. It can be said that wounding is an effective treatment to induce embryogenesis and the light spectrum may affect its efficiency.

PGRs are known to stimulate cell division and play an important role in the induction of SE [47]. The majority of previous studies have required auxins to induce SE in various plant species, while some reports have shown that cytokinins promote the formation of embryogenic cells with a role similar to that of auxins [48,49], for example, in orchids [5,15,50]. We used three concentrations of Kin (1, 1.5, and 2 mg $L^{-1}$), which had not previously been used for DSE in terrestrial orchids. Two concentrations of Kin (1.5 and 2 mg $L^{-1}$) resulted in positive effects on the DSE of *D. umberosa* (Table 1 and Figure 3). In other studies, the growth regulator kinetin has been used to induce protocorm formation in *Dactylorhiza majalis* [51,52]. The effect of these kinds of PGRs on embryogenesis has been reported on black iris [53], *Phalaenopsis* [54], and *Rhynchostylis retusa* [55]. Effective PGR treatments were 3 mg $L^{-1}$ TDZ on protocorm and 1 mg $L^{-1}$ NAA in nodal explant of *E. veratifolia* and 3 mg $L^{-1}$ TDZ and 2 mg $L^{-1}$ Kin on protocorm explant of *D. umberosa* (Table 1). The high concentrations of Kin and NAA (3 and 1 mg $L^{-1}$, respectively) had no positive effect on SE. It was reported that the exogenous addition of a high concentration of BA and Kin significantly inhibited SE formation in orchard grass [56]. Our results showed that TDZ had the highest efficiency for DSE than BA and NAA in *E. veratifolia*. This was consistent with the previous results in *Dendrobium* [50,57] and *Cymbidium* [33]. In general, we found that the application of TDZ and NAA on wounded explant was more effective for DSE in *E. veratifolia* and TDZ, Kin on wounded explant induced DSE in *D. umberosa*. There are only a few reports that mention the role of light spectra on SE induction in native orchid species. In the present experiment, the highest percentage of DSE and embryo germination rate in *E. veratifolia* was observed in protocorm explants exposed to R-Fr (100%) and then R (95%) spectra (Figure 6), while B spectra, W light, and D condition had less effect on embryo formation on explants (Figure 6a). This finding is in agreement with the results of previous studies on the initiation and development of SE by R and R-Fr spectra in *Araujia sericifera* [29], *Oncidium* [18], and *Rosa chinensis* [34]. In *E. veratifolia*, R and R-Fr spectra produced the highest rate of embryogenesis and proliferation of embryos (PLBs); although, the highest percentage of embryo proliferation was obtained under R-Fr light (Figures 5a and 7). The positive effects of the red light spectrum have been reported on the formation of 100% protocorms in orchids [58]. Consistently, R light has been reported to promote the induction and proliferation of SE in *Oncidium* [18]. Red light has been shown to affect plant reproduction through phytochrome, which in its active form increases the endogenous hormonal balance, increasing the amount of cytokinin in the tissues and counteracting the action of auxin [59]. In our study, in addition to R and R-Fr, G, R-B, B, W light and D conditions were used on explants to induce embryogenesis in *E. veratifolia*, but most of the explants produced a small number of embryos that also turned brown and

necrotic under these conditions during three to four weeks (Figure 9a,b). This shows that G, R-B, B spectra, W light, and D conditions delayed the germination of SE in *E. veratifolia*. In the current study, the maturation of embryos as well as the germination of embryos was significantly different in different light treatments. In some of the previous studies, it has been reported that callus formation was high in the blue spectrum, which proves that rapid cell division occurred, although the organized center of cell division required for primordia formation was reduced and growth was delayed [60]. Interestingly, we found that G light increased the formation and germination of SE in protocorm explants of *E. veratifolia* (Figures 4a and 6a). The effect of the G spectrum was significant on the percentage of explants that had embryogenesis and on the number of embryos (Figures 5a and 6a). A study on carrot embryogenesis showed that G light caused the highest SE compared to the other light spectra [61]. The highest PLBs formation, root formation, and shoot formation rate in *Cymbidium insigne* was reported under the G spectrum [62]. The G light may have active photoreceptors [63], but unlike red and blue light, green light photoreceptors have not been discovered yet [64]. Embryos germinated when sub-cultured in the hormone-free medium. In both orchids, the somatic embryos had advanced developmental stages under the B and R-B spectrum, which rapidly increased the size of the embryos (Figure 9d). This result shows the positive effects of the B spectrum on the development of PLBs and it is in agreement with the previous report on the promotive effects of the B spectrum on embryo size in carnation [24].

The results of this study show that the highest percentage of DSE (100%) was observed in the explants exposed to W light, while the dark condition had the lowest effect on embryo induction rate in *D. umberosa* (Figure 6b). Researchers have also reported that SE induction and promotion are reduced in darkness in olive [65], soybean [66], and *Coffea Arabica* [67]. The highest percentage of embryogenesis in *D. umberosa* was produced under the W light and B spectrum (Figure 6b). The B spectrum had an efficient effect on DSE and ISE in carnation [24]. The highest number of the torpedo, globular, and heart embryos was observed under the B spectrum; in comparison to the effect of other light treatments that had no embryo formation [24]. The B light can affect plant auxin content (specifically indoleacetic acid-IAA). Therefore, it can affect PGR and morphotomorphogenesis [68]. Similar to the previous species, in *D. umberosa*, the apical meristem of PLBs had the most development under R-B spectra and produced two to three-leaf primordia (Figure 2e). This issue shows the positive and promotive effects of the B spectrum. The result showed that when the G, R spectra and D conditions were used to induce embryogenesis in *D. umberosa*, most of the explants did not produce embryos or a high number of embryos died (Figure 9). The PLB with larger diameters were observed under the G spectrum and D conditions (Figure 3d,f), but these conditions were not suitable for their survival. In *D. umberosa*, the embryos germinated when they were transferred and cultured on the PGR-free basal medium, and the germination level of embryos in different light spectra was significantly different. After exposure to white light, the B spectrum increased the induction and germination of SE from the protocorm explant (Figures 6b and 8). The B spectrum is important for photomorphogenesis [68]. Consistent with this, it has been reported in *Oncidium* that the B spectrum increases differentiation and enzyme activities in embryos [18]. Previous studies have already mentioned the relationship between SE production and photo-equilibrium and the increase in SE formation by increasing photo-equilibrium. There is a higher photo-equilibrium under W light compared to the B spectrum [69]. Our results indicate that LEDs are a forward light source that increases DSE induction. Formation with high efficiencies of plant regeneration in both terrestrial orchids studied. We recommend the use of white light to produce the most PLBs on wounded protocorm explants of *D. umberosa* (Figure 8) and R-Fr light for producing the highest number of PLBs on wounded protocorm explants of *E. veratifolia* (Figure 7), both in a culture medium containing 3 mg L$^{-1}$ TDZ.

## 5. Conclusions

Some native species like *D. umberosa* have been exposed to extinction because of uncontrolled harvest from their natural habitats. Mass propagation of *E. veratrifolia* and *D. umberosa* is difficult and sometimes impossible by using conventional in vitro culture methods because of low seed germination, lack of suitable symbionts, slow vegetative growth, and absence of efficient methods for asexual reproduction. In the present study, for the first time, a successful method and an efficient in vitro propagation protocol were established for these species by using light spectra through DSE. This protocol will have a significant impact on commercial micropropagation and genetic resource conservation of these native orchids. Wounding on the protocorm explants of *D. umberosa* with the use of a medium containing 3 mg $L^{-1}$ TDZ caused a 94.1% increase in the number of PLBs under white light. In *E. veratrifolia*, the use of wounded protocorm explants in the medium containing 3 mg $L^{-1}$ TDZ under the R spectrum caused an increase in production by 98.8%.

**Author Contributions:** Conceptualization, H.N.B. and S.D.D.; Data curation, H.N.B.; Formal analysis, H.N.B. and S.D.D.; Investigation, H.N.B. and S.D.D.; Methodology, S.D.D.; Project administration, S.D.D.; Resources, S.D.D.; Software, H.N.B. and S.D.D.; Supervision, S.D.D.; Validation, S.D.D.; Writing—original draft, H.N.B.; Writing—review and editing, H.N.B., S.D.D. and K.V. All authors have read and agreed to the published version of the manuscript.

**Funding:** This research received no external funding.

**Data Availability Statement:** Not applicable.

**Acknowledgments:** The authors wish to thank the Orchid Breeding and Propagation Laboratory of the Horticultural Department, University of Tehran for the opportunity to carry out their research.

**Conflicts of Interest:** No potential conflict of interest were reported in this study.

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
