# Peer review of "Use of Light Spectra for Efficient Production of PLBs in Temperate Terrestrial Orchids"

_horticulturae, doi:10.3390/horticulturae9091007_

Round 1

Reviewer 1 Report

In the manuscript are described different experimental condition to obtain protocorm like bodies in the orchids D. umberosa and E. veratifolia. Different light quality, plant growth regulators, wounded and non-wounded plants have been employed.

In my opinion the manuscript needs to be improved and I hope the author can revise it carefully.

The reference number starts from 19 instead than 1. Please, sort out the list properly accordingly with the mdpi instructions.

As the title says, this research is based on different light quality. In the introduction the different light qualities and their role in the context of in vitro tissue culture is poorly treated. The authors should go into detail about this topic. Good information are provided in a recent review: Light and Plant Growth Regulators on In Vitro Proliferation. Cavallaro, V., Pellegrino, A., Muleo, R., Forgione, I. Plants, 2022, 11(7), 844.

In the results section, paragraph 3.1, lines 123 to 128 are a bit confusing. It is hard to connect the statistics with the table. The authors should explain better the concepts expressed in these lines.

In figure 1 and 2 PLBs grown under different light are shown without providing information on the PGRs treatment. The authors should improve the figures by adding arrows to indicate in detail the different parts of a PLB and somatic embryo.

The authors should explain better the procedure to obtain wounded and non-wounded PLBs even by providing detailed pictures.

In material and methods no number of seeds and explants is reported, as well as in the figures and in table1. The authors should include this information.

According to table 1, in E. veratrifolia TDZ is used only at 3 mg/L. Why?

And why TDZ is used only for protocorm and non in the other tissues? This should be explained. Also TDZ is not used in combination with others PGRs conversely than NAA used with BA. Why?

Why Kin in not used for E. veratrifolia?

In general, the logic on the combination of the treatments should be explained.

The discussion is redundant and often repeats sentences that fit more with the results. The authors should be more critical in respect to the light effects, trying to make some conclusion since thatf the light is also mentioned in the title. Is not sufficient state that the results obtained are in accordance with other researches. Similar thing for PGRs. The effects of PGRs in in vitro tissue culture are primary. So the different combinations oh these need to be discussed in order to motivate the results obtained.

In conclusion, in the present form the manuscript is not suitable to be published in Horticulture journal. I suggest major revision.

Minor comments:

Ensure to set in vitro in italics.

Line 125 andline change in and line.

Author Response

Dear Editor,

I would like to appreciate you and esteemed reviewers for your invaluable comments which gave us more opportunity to improve our manuscript. Following we addressed all the reviewers comments point by point. Hope our corrections and response letter make our paper suitable for publishing in Horticulturae prestigious journal.

Reviewer 1

The reference number starts from 19 instead than 1. Please, sort out the list properly accordingly with the mdpi instructions.

The reference number was edited.

As the title says, this research is based on different light quality. In the introduction the different light qualities and their role in the context of in vitro tissue culture is poorly treated. The authors should go into detail about this topic.

Thank you for your comment. The requested information was added.

In the results section, paragraph 3.1, lines 123 to 128 are a bit confusing. It is hard to connect the statistics with the table. The authors should explain better the concepts expressed in these lines.

It was changed according to the reviewers comments.

In figure 1 and 2 PLBs grown under different light are shown without providing information on the PGRs treatment. The authors should improve the figures by adding arrows to indicate in detail the different parts of a PLB and somatic embryo.

Thank you for your constructive comment. Figure were edited accordingly.

The authors should explain better the procedure to obtain wounded and non-wounded PLBs even by providing detailed pictures.

We appreciate your valuable comments. The following information were added to the manuscript.

Seeds were cultured on a modified FAST medium (MFAST) as basal medium containing macro and micro elements which was added to the materials and methods section. Orchids protocorms were formed when the seeds germinated. Then this explant was used as a protocorm explant. Protocorm was used without wounding in an intact way and without creating a cut. But, in the wounding treatment, the intact protocorm was cut and then used.

In material and methods no number of seeds and explants is reported, as well as in the figures and in table1. The authors should include this information.

Thank you for your comments. It was added.

According to table 1, in E. veratrifolia TDZ is used only at 3 mg/L. Why?

Thank you for your interesting question. We used different concentrations of this plant growth regulator in previous research on the tropical species like Phalaenopsis and observed the best somatic embryogenesis response at concentration of 3 mgl-1 according to the following paper:

GHAHREMANI, R.; DIANATI DAYLAMI, S.; MIRMASOUMI, M.; ASKARI, N.; and VAHDATI, K. (2021) "Refining a protocol for somatic embryogenesis and plant regeneration of Phalaenopsis amabilis cv. Jinan from mature tissues," Turkish Journal of Agriculture and Forestry: Vol. 45: No. 3, Article 11. https://doi.org/10.3906/tar-2004-107).

We studied different concentrations of TDZ on these two native species and obtained the best response in this concentration (unpublished results).

In the current research, we used hormone free culture media and also with growth regulators (also best concentration of TDZ) to compare them with the effects of light spectra. Our goal was to investigate whether light spectra can increase the effect of these growth regulators in the production of somatic embryos, or more importantly, can they replace growth regulators? The results showed that if the wounding treatment (cutting the explants) is used, especially on protocorm explants, the light spectra alone can be effective in somatic embryos (PLBs) production, and their effect is different based on the type of spectrum and the type of plant species. (Please see Figures 10 and 11).

And why TDZ is used only for protocorm and non in the other tissues? This should be explained. Also TDZ is not used in combination with others PGRs conversely than NAA used with BA. Why?

Thank you for your precise question. As mentioned above, in our previous studies, we investigated the effect of the most important growth regulators (BA, NAA, PBZ, KIN, TDZ, 2,4-D, ABA, etc.) on the embryogenesis of orchids, including these two native orchids. Some results of these studies have been published and some have not yet been published (For example, please see this paper: Shirin Moradi, Shirin Dianati Daylami, Mostafa Arab & Kourosh Vahdati (2017) Direct somatic embryogenesis in Epipactis veratrifolia, a temperate terrestrial orchid, The Journal of Horticultural Science and Biotechnology, 92:1, 88-97, DOI: 10.1080/14620316.2016.1228434). In the current research, we chose the best effective plant growth regulators treatments on each type of explants so that we investigated the effects of light spectra in interaction with them or without using them.

Thidiazol ureas or thidiazuron (TDZ) (that have been found that also exert cytokinin activity in various bioassay systems (Mok et al., 1986)) is up to 10,000 times more active than DPU, and more active than the adenine-type cytokinins. TDZ was reported to initiate direct PLB formation alone. Therefore, it is not recommended to use it together with other PGRs (Please see Chen JT, Chang WC (2002) Effects of tissue culture conditions and explant characteristics on direct somatic embryogenesis in Oncidium ‘Gower Ramsey’. Plant Cell Tissue Organ Cult 69:41–44.- and Wang CX, Tian M (2014) Callus-mediated and direct protocorm-like body formation of Bletilla striata and assessment of clonal fidelity using ISSR markers. Acta Physiol Plant 36:2321–2330. And Murch SJ, Saxena PK (2001) Molecular fate of thidiazuron and its effects on auxin transport in hypocotyls tissues of Pelargonium × hortorum Bailey. Plant Growth Regul 35:269–275.)

Protocorms generally have high potential for producing a whole plantlet that make it suitable explants for rapidly proliferation and regeneration into PLBs (Please see Naing AH, Chung JD, Park IS, Lim KB. Efficient plant regeneration of the endangered medicinal orchid, Coelogyne cristata using protocorm-like bodies. Acta Physiologiae Plantarum. 2011 May;33:659-66.- and Bustam S, Sinniah UR, Kadir MA, Zaman FQ, Subramaniam S. Selection of optimal stage for protocorm-like bodies and production of artificial seeds for direct regeneration on different media and short term storage of Dendrobium Shavin White. Plant Growth Regulation. 2013 Apr;69:215-24.).We have found that use of this type of explant (protocorm) in combination with TDZ produces the best results for achieving direct somatic embryogenesis (direct PLBs formation). Of course, our main goal in the present study was to compare the results of these findings with the effect of light spectra alone on direct somatic embryogenesis and PLBs production.

In some studies, the combination of BA and NAA have been used for the production of PLBs and it has become well known as a successful plant growth regulator in plant cell, tissue and organ culture, for example please see 1. Li, Z.Y. and Xu, L., 2009. In vitro propagation of white-flower mutant of Rhynchostylis gigantea (Lindl.) Ridl. through immature seed-derived protocorm-like bodies. J. Hort. Forest, 1, pp.93-7.- Baker, A., Kaviani, B., Nematzadeh, G. and Negahdar, N., 2014. Micropropagation of Orchis catasetum–a rare and endangered orchid. Acta Scientiarum Polonorum Hortorum Cultus13(2), pp.197-205.) The use of these PGRs is more economic compared to TDZ, so if the interaction of light and their effective concentrations would produce a better result than TDZ, it would encourage us to use them. But the results were not significant compared to TDZ, especially on protocorm as a high potential explant.

Why Kin in not used for E. veratrifolia?

Thank you for your interesting question. In our previous studies, it was found that KIN does not have a significant effect on direct embryogenesis in E. veratrifolia. As can be seen in Table 1, only half of the protocorm explants of D. umberoza had direct embryogenesis by using KIN.
The discussion is redundant and often repeats sentences that fit more with the results. The authors should be more critical in respect to the light effects, trying to make some conclusion since that the light is also mentioned in the title. Is not sufficient state that the results obtained are in accordance with other researches. Similar thing for PGRs. The effects of PGRs in in vitro tissue culture are primary. So the different combinations on these need to be discussed in order to motivate the results obtained.
We are grateful to the reviewer who is trying to improve our paper by this comment. According to the reviewer’s opinion, the manuscript was corrected.

Ensure to set in vitro in italics., Line 125 andline change in and line.

Corrected accordingly. 

Reviewer 2 Report

Dear authors, thanks a lot for such work manipulating the tissue culture of some examples of Orchids. Would you please follow the following points to improve the work.

Abstract: please mention the full name of the following abbreviation of “PGRs, DSE, PLBs and TDZ and all other phytohormones”.

Abstract: in line 23, please capitalized “R-Fr” to be “R-FR”.

Keywords: please add the full botanical name of “Epipactis veratrifolia and D. Umberosa”, to the list of keywords.

Introduction: please arrange the references in the text. (you began with 19, 18, 56, 49, 36, ….) is it logic!.

Materials and methods: I line 79, make the letter small in “Umberosa” as it is species name.

Discussion: all references in text were not arranged.

Supplementary materials are not provided

The plagiarism percentage was 39% , please try to reduce it

Minor editing of English language required

Author Response

Dear Editor,

I would like to appreciate you and esteemed reviewers for your invaluable comments which gave us more opportunity to improve our manuscript. Following we addressed all the reviewers comments point by point. Hope our corrections and response letter make our paper suitable for publishing in Horticulturae prestigious journal.

Reviewer 2

Abstract: please mention the full name of the following abbreviation of “PGRs, DSE, PLBs and TDZ and all other phytohormones”.

Thank you for your comment. The full names were added.

Abstract: in line 23, please capitalized “R-Fr” to be “R-FR”.

It was changed accordingly.

Keywords: please add the full botanical name of “Epipactis veratrifolia and D. Umberosa”, to the list of keywords.

It was added.

Introduction: please arrange the references in the text. (you began with 19, 18, 56, 49, 36, ….) is it logic!.

It was changed accordingly.

Materials and methods: I line 79, make the letter small in “Umberosa” as it is species name.

Corrected.

Discussion: all references in text were not arranged.

Thank you for your comment. Done.

Supplementary materials are not provided

We didn’t have supplementary materials in this manuscript.

The plagiarism percentage was 39% , please try to reduce it.

Thank you for your precise comment. We did our best to reduce it to less than 30%.

Reviewer 3 Report

Lines 15 and 21, :  Must explain the abbreviations when first mentioned in the manuscript.

Lines 17-18: D. umberosa and E. veratifolia must be replaced with entire scientific species name when first mentioned in the manuscript abstract.

Line 79: D. Umberosa should be replaced with  D. umberosa

Line 79: “Capsules of these plants were collected”. It should be mentioned the capsules maturity stage.

Line 84: what you mean “wounded and non-wounded” protocorms

Line 85: “were prepared for the experiment”. What was the preparation procedure.

Line 101: “peak at 400-101 700 nm”. This is a range not a peak.

Line 123: “Three weeks after culture, initiation of embryos was observed on the explants” should be replaced with “Three weeks after placing the explants in the growth medium, initiation of embryos was observed”

Line 125: “andlight” add space between the words

Line 126 and 136: Τhe treatments showing a statistically significant difference for P<0.05 and P<0.01) should be specified.

Line 128: Figures should be renumbered according to their order of reference in the text

Line 134: ” Three weeks after culture” should be replaced with “Three weeks after placing the explants in the growth medium”.

Fig 3,4,5,6,7: the horizontal axes will need to be titled. Standard deviations over many columns are not symmetrical. Vertical axis in all three figures is better to have the same scale in order the measurements become comparable. I same cases (Fig. 3) the presented SD doesn’t

Lines 141-143: “Maximum rate for embryo formation (100, 50 and 25% of DSE) on protocorms of D. umberosa, was obtained on 3 mgl-1 TDZ, 2 mgl-1 and 1.5 mgl-1 Kin, respectively” The maximum rate of embryo formation of DSE should be represented from one value in this case 100% on 3 mgl-1 TDZ.

Fig 3 and 6: In some cases the standard deviation combined with the mean value presented in the graph does not justify the absence of a statistically significant difference between treatments. For example, in Fig 3, column named b (treatment with 3 mg/l) and column named R (treatment with 0 mg/l) have very low SD, however signed with the same letter "h". The same case concerns the two columns named W as well as the two columns named R (treatments with wounded and non-wounded protocorms) in Fig. 6 signed with the same letter "a" and "c" respectively.

Line 154: “wase” should be replaced with “was”

Fig 7: The first column is signed with f5 to denote statistically significant differences with the other columns. Is this correct?

Fig 5,6,7,8,9,10: Correction in the figures caption,  the horizontal axis shows the light treatment not the vertical one.

Fig 8,9,10: In figures caption is mentioned the expression “total means” ?

Line 198:” for example, [52] reported”. This is not a correct expression

Line 290: In the conclusions, general references are made to the objectives of the work. Instead, the most important results should be reported

Minor editing of English language required

Author Response

Dear Editor,

I would like to appreciate you and esteemed reviewers for your invaluable comments which gave us more opportunity to improve our manuscript. Following we addressed all the reviewers comments point by point. Hope our corrections and response letter make our paper suitable for publishing in Horticulturae prestigious journal.

Lines 15 and 21:  Must explain the abbreviations when first mentioned in the manuscript

Thank you for your comments that improved our manusctip. Done.

Lines 17-18: D. umberosa and E. veratifolia must be replaced with entire scientific species name when first mentioned in the manuscript abstract.

Done.

Line 79: D. Umberosa should be replaced with D. umberosa

Done.

Line 79: “Capsules of these plants were collected”. It should be mentioned the capsules maturity stage?

Done.

 Line 84: What you mean “wounded and non-wounded” protocorms (line 84)

Thank you for your interesting question. Non-wounded means intact and uncut protocorms and wounded means cut protocorms to investigate the wound effect on this explant. Image was added to Figure 1.

Line 85: “were prepared for the experiment”. What was the preparation procedure?

We appreciate your constructive comment. The description was completed and the figure was added.

Line 101: “peak at 400-101 700 nm”. This is a range not a peak.

Corrected.

Line 123: “Three weeks after culture, initiation of embryos was observed on the explants” should be replaced with “Three weeks after placing the explants in the growth medium, initiation of embryos was observed”

Done.

Line 125: “andlight” add space between the words

Done.

Line 126 and 136: Τhe treatments showing a statistically significant difference for P<0.05 and P<0.01) should be specified.

Done.

Line 128: Figures should be renumbered according to their order of reference in the text??

All done. Figure . 1 is related to the materials and work method was mentioned in that part.

Line 134: ” Three weeks after culture” should be replaced with “Three weeks after placing the explants in the growth medium”.

Done.

Fig 3,4,5,6,7: the horizontal axes will need to be titled. Standard deviations over many columns are not symmetrical. Vertical axis in all three figures is better to have the same scale in order the measurements become comparable. I same cases (Fig. 3) the presented SD doesn’t

Thank you for your comment. All were corrected accordingly.

Lines 141-143: “Maximum rate for embryo formation (100, 50 and 25% of DSE) on protocorms of D. umberosa, was obtained on 3 mgl-1 TDZ, 2 mgl-1 and 1.5 mgl-1 Kin, respectively” The maximum rate of embryo formation of DSE should be represented from one value in this case 100% on 3 mgl-1 TDZ.

Done.

Fig 3 and 6: In some cases the standard deviation combined with the mean value presented in the graph does not justify the absence of a statistically significant difference between treatments. For example, in Fig 3, column named b (treatment with 3 mg/l) and column named R (treatment with 0 mg/l) have very low SD, however signed with the same letter "h". The same case concerns the two columns named W as well as the two columns named R (treatments with wounded and non-wounded protocorms) in Fig. 6 signed with the same letter "a" and "c" respectively.

Done.

Line 154: “wase” should be replaced with “was”

Done.

Fig 7: The first column is signed with f5 to denote statistically significant differences with the other columns. Is this correct?

Done.

Fig 5,6,7,8,9,10: Correction in the figures caption,  the horizontal axis shows the light treatment not the vertical one.

Done.

Fig 8,9,10: In figures caption is mentioned the expression “total means” ?

Done.

Line 198:” for example, [52] reported”. This is not a correct expression

Done.

Line 290: In the conclusions, general references are made to the objectives of the work. Instead, the most important results should be reported

Done.

Reviewer 4 Report

The manuscript submitted by Boldaji and collaborators reported an optimized protocol for in vitro massive propagation of two native orchid species. Due to the conservation status of the two species, the study is relevant as an alternative way of propagation. The document is in preliminary form still; authors should make some improvements before considering this manuscript for publication.

Here are some issues detected:

Abstract

L15: DSE? Not use uncommon abbreviations not declared previously.  In fact, this abbreviation was not declared anywhere in the document.

L17: D. umberosa? Not use uncommon abbreviations not declared previously.

L18: E. veratifolia? Not use uncommon abbreviations not declared previously.

L21, L23: Please, check spelling for the units mg L-1, in this section, and throughout the manuscript.

Introduction

L73: No need for the punctuation mark hyphen in the words in vitro

M&M

L84, L99: Please provide more information about the wounding treatment given to the protocorms.

L97: Not Capital letters for the following words: crown, node, leaf, and protocorm.

L108: Binocular? Binocular microscope? Confirm it, please.

L09: FAST hormone-free medium or hormone-free FAST medium?

Provide more information for all chemicals used in this study (trademark, origin country).

Results

The descriptions of the results are too short. Is it possible abundance on them a bit more?

Figures and tables should be included in sections 3.1 and 3.2 and not in an independent section.

When figures are cited, there is no need of a period after the word figure.

Why figures 1 and 2 are not cited in the text?

Figures 3, 4, 9, 10: Y-axis: PLB no.? PLB number?

A suggestion: All the graphs have a good resolution, so figures with content/information related might integrate into composite figures instead of single figures (v.g., Figures 3 and 4).

Figures 1 and 2: Scales for SE structures: the scale is indicated only for one panel (fig. 1 h) (fig. 2, a). Is the same scale for the rest of panels?

Figure captions: Check grammar in some figure captions. Figure 3 and 4: a period is missing before the word different.

Figures 3-10 captions: Indicate if the values are means ± SD or SE.

Figures 3-10 captions: Different letters show significant differences among treatments according…? (P≤0.05), Duncan’s multiple range tests?

Figure 11: Scale is missing.

Minor editing of English language required.

Author Response

We appreciate your comments for improving our manuscript. The text was corrected according your invaluable comments.

L15: DSE? Not use uncommon abbreviations not declared previously.  In fact, this abbreviation was not declared anywhere in the document.

Done.

L17: D. umberosa? Not use uncommon abbreviations not declared previously.

Done.

L18: E. veratifolia? Not use uncommon abbreviations not declared previously.

Done.

L21, L23: Please, check spelling for the units mg L-1, in this section, and throughout the manuscript.

 Done.

Introduction

L73: No need for the punctuation mark hyphen in the words in vitro

 In vitro used in italic.

M&M

L84, L99: Please provide more information about the wounding treatment given to the protocorms.

Done.

L97: Not Capital letters for the following words: crown, node, leaf, and protocorm.

Done.

L108: Binocular? Binocular microscope? Confirm it, please.

Done.

L09: FAST hormone-free medium or hormone-free FAST medium?

Done.

Provide more information for all chemicals used in this study (trademark, origin country).

Done.

Results

The descriptions of the results are too short. Is it possible abundance on them a bit more?

Done.

Figures and tables should be included in sections 3.1 and 3.2 and not in an independent section.

These are prepared based on horticulturae format.

When figures are cited, there is no need of a period after the word figure.

Done.

Why figures 1 and 2 are not cited in the text?

Done.

Figures 3, 4, 9, 10: Y-axis: PLB no.? PLB number?

Done.

A suggestion: All the graphs have a good resolution, so figures with content/information related might integrate into composite figures instead of single figures (v.g., Figures 3 and 4).

Done.

Figures 1 and 2: Scales for SE structures: the scale is indicated only for one panel (fig. 1 h) (fig. 2, a). Is the same scale for the rest of panels?

Done.

Figure captions: Check grammar in some figure captions. Figure 3 and 4: a period is missing before the word different.

Done.

Figures 3-10 captions: Indicate if the values are means ± SD or SE.

Done.

Figures 3-10 captions: Different letters show significant differences among treatments according…? (P≤0.05), Duncan’s multiple range tests?

Done.

Figure 11: Scale is missing.

Done.

Round 2

Reviewer 4 Report

The authors have addressed all the issues made in the submitted original manuscript. At this point, I have no more comments. The manuscript is ready for publication.